# Transfer Learning and Neural Network-Based Approach on Structural MRI Data for Prediction and Classification of Alzheimer’s Disease

**DOI:** 10.3390/diagnostics15030360

**Published:** 2025-02-04

**Authors:** Farideh Momeni, Daryoush Shahbazi-Gahrouei, Tahereh Mahmoudi, Alireza Mehdizadeh

**Affiliations:** 1Department of Medical Physics, School of Medicine, Isfahan University of Medical Sciences, Isfahan 81746-73461, Iran; momenifarideh777@gmail.com; 2Department of Medical Physics and Engineering, School of Medicine, Shiraz University of Medical Sciences, Shiraz 71348-14336, Iran; t.mahmoudi94@gmail.com (T.M.); mehdizade@sums.ac.ir (A.M.)

**Keywords:** structural MRI, deep learning, transfer learning, Alzheimer’s disease, mild cognitive impairment

## Abstract

**Background:** Alzheimer’s disease (AD) is a neurodegenerative condition that has no definitive treatment, and its early diagnosis can help to prevent or slow down its progress. Structural magnetic resonance imaging (sMRI) and the progress of artificial intelligence (AI) have significant attention in AD detection. This study aims to differentiate AD from NC and distinguish between LMCI and EMCI from the other two classes. Another goal is the diagnostic performance (accuracy and AUC) of sMRI for predicting AD in its early stages. **Methods:** In this study, 398 participants were used from the ADNI and OASIS global database of sMRI including 98 individuals with AD, 102 with early mild cognitive impairment (EMCI), 98 with late mild cognitive impairment (LMCI), and 100 normal controls (NC). **Results**: The proposed model achieved high area under the curve (AUC) values and an accuracy of 99.7%, which is very remarkable for all four classes: NC vs. AD: AUC = [0.985], EMCI vs. NC: AUC = [0.961], LMCI vs. NC: AUC = [0.951], LMCI vs. AD: AUC = [0.989], and EMCI vs. LMCI: AUC = [1.000]. **Conclusions:** The results reveal that this model incorporates DenseNet169, transfer learning, and class decomposition to classify AD stages, particularly in differentiating EMCI from LMCI. The proposed model performs well with high accuracy and area under the curve for AD diagnostics at early stages. In addition, the accurate diagnosis of EMCI and LMCI can lead to early prediction of AD or prevention and slowing down of AD before its progress.

## 1. Introduction

Alzheimer’s disease (AD) is a progressive neurodegenerative disorder that manifests clinically as memory loss, visuospatial difficulties, and changes in personality and behavior [1,2]. The early and accurate diagnosis of AD is crucial for slowing disease progression and enabling timely intervention [3,4]. Significant advances have been made in detecting AD pathology using cerebrospinal fluid (CSF) biomarkers [5,6,7], positron emission tomography (PET) amyloid imaging [8,9], and tau imaging [10,11,12]. However, these diagnostic approaches are often limited to research settings.

Generally speaking, the diagnosis of AD relies on skilled neurologists who assess patient history, conduct objective cognitive evaluations like the Mini-Mental State Examination (MMSE) or neuropsychological tests [13], and utilize structural MRI (sMRI) to identify brain changes suggestive of AD [6]. Various biomarkers, including amyloid and tau proteins [14], CSF [15], and plasma markers [16,17,18] are being investigated to facilitate early detection, which holds the potential for promoting timely intervention and prevention [19]. Mild cognitive impairment (MCI) is recognized as an intermediate stage between normal aging and AD. It is considered a precursor to AD, particularly when associated with memory deficits and impaired judgment.

Although the exact cause of AD remains unclear and no definitive treatment currently exists, early detection is essential for slowing its progression. However, distinguishing between stable MCI patients (who do not progress to AD) and MCI converters (who eventually develop AD) remains a significant challenge. Magnetic resonance imaging (MRI) as a non-invasive method plays an important role in routine clinical evaluations and is recognized as a key biomarker for monitoring AD progression [20,21,22]. Since 2013, deep learning (DL) has garnered significant attention in the diagnosis and treatment of different abnormalities, in particular after 2017 [23]. Many studies have focused on structural brain changes to identify atrophy associated with AD and its prodromal stages, often employing voxel-based methods. These methods utilize values of voxel intensity from functional MRI (fMRI) methods, and approximately 70% of such works conduct whole-brain analyses [24]. The primary advantage of whole-brain analysis lies in its ability to integrate spatial data, enabling the acquisition of three-dimensional (3D) information from FMRI. Therefore, the voxel-based approach has a key drawback: it increases data dimensionality and computational complexity [25]. Despite these challenges, the voxel-based method has been widely applied in various studies [26,27].

Deep learning models based on two-dimensional (2D) data involve fewer parameters and require shorter training times than those using 3D. To minimize the number of parameters, many studies have adopted slice-based approaches, extracting 2D slices from 3D brain imaging. This approach may result in information loss, as reducing volumetric data to 2D representations omits the inherent 3D structure of brain tissue [24,25]. Consequently, a slice-based technique cannot provide comprehensive whole-brain information. Many studies have explored unique methods for extracting 2D slices from 3D brain images to address this limitation, while others have relied on standard projections along the axial, sagittal, and coronal views [28,29].

Over the years, researchers have explored various methods, including traditional machine learning algorithms and advanced DL techniques, for the individualized early diagnosis of AD [24]. In recent years, DL approaches, particularly convolutional neural networks (CNNs), have demonstrated considerable success in image classification and computer vision tasks [3]. To develop the model, the researchers utilized sagittal, coronal, and axial MR images from the dataset, each corresponding to a specific brain location, to train a 2D CNN model. Initially, individual base classifiers were trained using slices from a single axis (sagittal, coronal, or axial). Subsequently, an ensemble of classifiers was formed for each axis, and the base classifiers that demonstrated high performance were selected using the required data. These base classifiers were combined to create a final enhanced classifier ensemble incorporating information from all three axes. The enhanced lightweight CNN (EL-CNN) model had an accuracy of 62.0% in distinguishing patients with MCI who were likely to convert to AD from those in stable conditions [30]. In a study, Liu et al. [3] introduced a multi-model DL framework leveraging CNNs for the dual tasks of automatic hippocampal segmentation and AD classification using sMRI data. The framework was evaluated on baseline T_1_-weighted MRI data from the AD Neuroimaging Initiative (ADNI) database including 97 AD, 233 mild cognitive impairment (MCI), and 119 normal control (NC) subjects. Their proposed method showed a dice similarity coefficient of 87.0% for hippocampal segmentation. For classification tasks, it demonstrated an accuracy of 88.9% and an AUC (area under the curve) of 92.5% in differentiation AD from NC subjects. Additionally, it obtained 76.2% accuracy and 77.5% AUC for differentiating MCI from NC [3].

In another work, Bae et al. [31] modified a CNN model to predict which individuals with mild cognitive impairment (MCI) would convert to AD and which would not convert (MCI-NC). This model was initially trained on sMRI from healthy individuals and those with AD to perform a classification task distinguishing NC from AD. This served as the source task for transfer learning. The knowledge acquired during the NC versus AD classification was transferred to the target task of distinguishing MCI-C from MCI-NC. Subsequently, the model was fine-tuned using sMRI from MCI patients to extract features specific to AD conversion. This approach showed an accuracy of 82.4% in predicting the progression from MCI to AD [31].

Consequently, Maha et al. [27] introduced a class decomposition transfer learning (CDTL) approach that leverages pre-trained models like VGG19 and AlexNet combined with an entropy-based technique to detect AD from sMRI. They have evaluated the robustness of the CDTL method in detecting cognitive decline associated with AD across various ADNI cohorts, aiming to determine if comparable classification accuracy could be achieved for two or more cohorts. Impressively, the proposed model demonstrated state-of-the-art performance, achieving an accuracy of 91.45% in predicting the conversion from MCI to AD.

All mentioned studies have been conducted for binary (AD compared to NC) or triple (AD, NC, MCI) classification. However, distinguishing AD from NC and differentiation between LMCI and EMCI from the other two classes is very important. Also, the DenseNet169 was performed, and ImageNet was trained on it in the literature. In this study, the DenseNet model was accomplished on MRI data and adding the layers to achieve more reasonable results. The main goal of this work is to differentiate AD from NC and distinguish between LMCI and EMCI from the other two classes. Another goal is the diagnostic performance (accuracy and AUC) of sMRI for predicting AD in its early stages.

## 2. Materials and Methods

### 2.1. Dataset

This study utilized an MRI dataset on AD classification from two prominent global databases: the AD Neuroimaging Initiative (ADNI) database (Keck School of Medicine of USC, Los Angeles, CA, USA) and the Open Access Series of Imaging Studies (OASIS), Washington University, School of Medicine in St. Luis, USA. Alzheimer’s disease neuroimaging initiative, established in 2003 as a public-private partnership under the leadership of Principal Investigator Michael W. Weiner, MD, aims to evaluate whether serial MRI, PET, other biological markers, and clinical/neuropsychological assessments can collectively track the progression of mild cognitive impairment (MCI) and early AD. Open Access Series of Imaging Studies is a project dedicated to making brain neuroimaging datasets freely accessible to the scientific community, to advance discoveries in basic and clinical neuroscience.

A total of 398 participants from the ADNI and OASIS database of sMRI (T_1_-weighted) MPRAGE at 1.5 T, were randomly selected including 98 individuals with AD, 102 with early mild cognitive impairment (EMCI), 98 with late (LMCI) and 100 normal controls (NC). The total number of images was 32,160, of which 70% were used for training, 20% for validation, and 10% for testing. Baseline T_1_-weighted MRI data and pre-processed images were used. The demographic and clinical information are summarized in Table 1, where the clinical dementia rating (CDR) and mini-mental state examination (MMSE) scores are provided as key indicators. All MR images were acquired using 1.5 T scanners by the ADNI acquisition protocol. Handling high-resolution 3D images poses significant challenges, including the large data volume, high computational complexity, and extensive storage requirements.

### 2.2. Preprocessing

To prepare the dataset for optimal performance in the proposed model, several preprocessing steps were applied. These steps ensured consistent input quality, improved model generalization, and mitigated overfitting. The preprocessing pipeline is detailed as follows: First, all input images were resized to a fixed resolution of 224 × 224 pixels to align with the input dimensions required by the DenseNet169 model. This resizing step ensured uniformity in input size, which is critical for batch processing and efficient training. Second, the pixel intensity values of the images were normalized to the range [0, 1] by dividing each pixel value by 255. This normalization step not only ensured numerical stability during training but also facilitated faster convergence by bringing input data into a consistent scale. Third, to improve model robustness and increase the diversity of the training data, several augmentation techniques were applied. These augmentations simulated variations that could occur in real-world clinical scenarios and included the following: Images were randomly rotated up to ±30 degrees to simulate angular variations in image acquisition. Both horizontal and vertical flipping were applied randomly to introduce variations in orientation. Random zooming of up to 20% was employed to mimic differences in field of view during image capture. These augmentation techniques not only enriched the dataset but also acted as a regularization mechanism to reduce the risk of overfitting, especially given the limited size of medical imaging datasets. Fourth, the data augmentation process was implemented using the ImageDataGenerator module from TensorFlow. This module dynamically generated augmented image batches during training, ensuring that the model was exposed to new variations in every epoch without requiring additional storage for augmented images. Fifth, the dataset was split into training, validation, and test sets to ensure a fair evaluation of the model. The training set was augmented, while the validation and test sets were left unaltered to evaluate the model’s performance on unseen, non-augmented data. By combining resizing, normalization, and augmentation, the preprocessing pipeline ensured that the dataset was well-prepared for deep learning-based classification, leading to improved generalization and robustness in the proposed method.

### 2.3. Model Architecture

As shown in Figure 1, DenseNet169 was employed as the base model for feature extraction due to its densely connected architecture, which ensures efficient gradient flow and feature reuse across layers. This pre-trained network was fine-tuned to adapt to the domain-specific characteristics of MRI images. By freezing the pre-trained layers during initial training, the model retained general visual features learned from ImageNet, while the added fully connected layers were trained to focus on the nuances of the MRI data. The modified DenseNet169 model was extended with custom layers, including batch normalization, dense layers, dropout layers for regularization, and a Softmax activation layer to output probabilities for the four target classes. This architecture enabled precise classification, as demonstrated by the high performance across metrics such as AUC and loss. The architecture of the model is illustrated as follows:

#### 2.3.1. Base Feature Extractor

The base feature extractor includes DenseNet169 with frozen pre-trained weights and its output (feature map of shape (7, 7, 1664) (7, 7, 1664)).

#### 2.3.2. Custom Layers

Custom layers include: (a) flattening the feature map to a one-dimensional vector, (b) fully connected (Dense) layers with ReLU activations, (c) dropout layers (rate = 0.5) to prevent overfitting, (d) batch normalization layers for stable and faster training, and (e) output layer: a dense layer with 4 neurons and Softmax activation for multi-class classification. The model was compiled using the Adam optimizer with a learning rate of 0.001 and the categorical cross-entropy loss function. The AUC was used as the primary evaluation metric.

### 2.4. Training Procedure

The training process consisted of a batch size of 128, several epochs: initially set to 100 (with early stopping to halt training if validation AUC did not improve for 15 consecutive epochs), and callbacks: early stopping (patience = 15), model checkpointing to save the best-performing model based on validation AUC. The training process was conducted on an NVIDIA GPU for faster computation.

### 2.5. Evaluation

The model was evaluated on a separate test set using the following metrics: loss (categorical cross-entropy) and AUC evaluated for multi-class classification. In addition, the model’s performance was visualized using training and validation loss curves and training, and validation AUC curves. This structured methodology ensures a robust and reproducible framework for classifying Alzheimer’s-related MR images.

## 3. Results

The performance of the proposed model was evaluated comprehensively using the test dataset, leveraging both quantitative metrics and qualitative visualizations. The DL approach utilized a pre-trained DenseNet169 model as the feature extractor, capitalizing on its robust feature representation learned from the ImageNet dataset. This transfer learning strategy significantly improved the model’s ability to distinguish between various stages of (AD, EMCI, LMCI) and NC based on MR images.

### 3.1. Training and Validation Metrics

The categorical cross-entropy loss and the AUC were used to see how well the model learned from training data.

As shown in Figure 2a, the training loss curve indicated a steady decline over the epochs, reflecting the model’s increasing ability to minimize errors in the training data. The validation loss initially followed the training loss but plateaued toward the later epochs, suggesting the model successfully avoided overfitting and retained good generalization capabilities. This stabilization is crucial in medical imaging tasks, where overfitting can lead to poor performance on unseen test data. AUC measures the model’s ability to distinguish between classes. Both the training and validation AUC curves consistently improved over epochs. Validation AUC reached a plateau close to the training AUC, indicating the model generalized well to unseen validation data without significant overfitting. The AUCs for training and validation datasets are shown in Figure 2b.

### 3.2. ROC Curves and AUC Values

The ROC curves and corresponding AUC values provide a detailed analysis of the model’s discriminative ability for each binary classification task (AD vs. NC, EMCI vs. LMCI, EMCI vs. NC, LMCI vs. NC, LMCI vs. AD). These metrics are particularly important in medical imaging, where carrying out a balance between sensitivity (recall) and specificity is critical. The ROC curves for each binary class are plotted to visualize the trade-off between true positive rates (sensitivity) and false positive rates (1-specificity) as indicated in Figure 3. A near-perfect receiver operating characteristic (ROC) curve approaches the top-left corner of the graph, indicating excellent performance.

As shown in Table 2, different AUC values were obtained in terms of area under the ROC curve for binary classification. The model achieved high AUC values for all four classes: NC vs. AD: AUC = [0.985], EMCI vs. NC: AUC = [0.961], LMCI vs. NC: AUC = [0.951], LMCI vs. AD: AUC = [0.989], EMCI vs. LMCI: AUC = [1.000]. These results highlight the model’s capability to differentiate between disease stages, with particularly high confidence in distinguishing AD and NC classes. However, a slight decrease in AUC for EMCI and LMCI classes suggests that these stages due to their clinical similarity, remain challenging to classify.

The confusion matrix and the values of the F1-score for each class were also obtained. According to Table 3, the model achieved strong performance across all classes, with F1-scores of 0.94, 0.99, 0.99, and 0.98 for AD, NC, EMCI, and LMCI, respectively. The F1-score is a metric that provides a balanced measure of a model’s performance by combining precision and recall into a single value. It is particularly useful for imbalanced datasets where one class may have more samples than the others [32]. An F1-score close to 1 indicates excellent precision and recall, meaning the model performs well in identifying true positives and avoiding false positives. In this study, the F1-scores for all classes were high, ranging from 0.94 (for AD) to 0.99 (for NC and EMCI). This indicates that the model is highly effective in classifying MRI images into their respective categories with minimal trade-offs between precision and recall. The highest classification accuracy was observed for the NC and EMCI classes, with minimal misclassifications. Misclassifications were more common between the AD and NC classes, likely due to subtle similarities in certain features of these groups. For example, 100 samples of AD were incorrectly classified as NC, and 14 samples of NC were incorrectly classified as AD. For LMCI, a small number of misclassifications occurred: Four LMCI samples were classified as AD, and 14 LMCI samples were classified as NC. These results indicate the model’s robustness in distinguishing between cognitively distinct classes while highlighting minor areas for improvement in reducing misclassifications between closely related groups like AD and NC or LMCI and NC. These values and the prediction graph of each class compared to each other are presented in Figure 4.

## 4. Discussion

Using a pre-trained network for image classification offers several advantages as follows:

1. Time and resource efficiency: Training a neural network from scratch can be time-consuming and computationally expensive. By leveraging a pre-trained model, the initial stages of training and fine-tuning of the model for specific tasks can be skipped by saving both time and computational resources.

2. Improved performance with limited data: Pre-trained networks are usually trained on large datasets (e.g., ImageNet), enabling them to learn a wide range of features. This means that even with a smaller dataset for the task, the pre-trained model can perform better than a model trained from scratch.

3. Better generalization: Since pre-trained networks have already been exposed to various images and features, they are often more capable of generalizing to new, unseen data. This makes them highly effective in diverse applications and reduces the risk of overfitting on limited datasets.

4. Transfer learning: Pre-trained models facilitate transfer learning, where the learned features from one task (e.g., object recognition) can be reused for another task (e.g., medical image classification), improving the training efficiency for new domains.

5. State-of-the-art accuracy: Many pre-trained networks are based on the latest advancements in deep learning, which have been tested and optimized for image classification tasks. This ensures a benefit from state-of-the-art performance in terms of accuracy and robustness.

In summary, using pre-trained networks is a powerful approach that leads to faster, more efficient, and more accurate image classification, especially when data or resources are limited.

The results of this study demonstrate the robustness and efficiency of the proposed DL model, based on DenseNet169, in classifying AD and its prodromal stages. The high AUC values achieved across all classes underscore the model’s ability to effectively differentiate between cognitively normal individuals and those in various stages of cognitive decline. Specifically, the model achieved an AUC of 0.985 for NC vs. AD, 0.961 for EMCI vs. NC, 0.951 for LMCI vs. NC, 0.989 for LMCI vs. AD, and a perfect 1.000 for EMCI vs. LMCI, respectively. These results highlight the model’s strength in distinguishing between disease stages, although clinical similarities between certain stages, such as EMCI and LMCI, present classification challenges.

### 4.1. Comparison with Previous Studies

Variations in datasets, data preprocessing strategies, dimensionality reduction techniques, and evaluation criteria have made it difficult to compare the findings of previous studies. Nevertheless, despite differences in experimental configurations, the outcomes of various methods remain comparable. The model was evaluated against the approaches outlined in Table 4, relying on the performance metrics reported in their respective publications.

These studies were chosen due to their frequent citation in the literature for similar or closely related tasks. As presented in Table 4, this proposed model surpasses earlier methods in differentiating EMCI from LMCI achieving an accuracy of 99.7%.

In this study, the methodology provides several key advantages over previous approaches. The use of transfer learning and class decomposition in the base classifier effectively addresses the challenge of limited labeled data, while also enhancing model performance by simplifying the process of learning class boundaries. Transfer learning was implemented to leverage knowledge gained from distinguishing AD from NC. Given that the differences between EMCI and LMCI are expected to be subtler than those between AD and NC, models trained on data from AD and NC prove highly effective for separating EMCI and LMCI. Consequently, the pre-trained model achieves outstanding performance in distinguishing EMCI from LMCI.

### 4.2. Strengths of the Proposed Approach

In this study, the key point is the integration of DenseNet169 as a feature extractor. By leveraging pre-trained weights and fine-tuning them for domain-specific tasks, the model capitalized on the architectural advantages of densely connected layers, such as efficient gradient propagation and feature reuse. The entropy-based slice selection ensured the retention of the most informative MRI regions, thereby reducing computational complexity without compromising diagnostic accuracy. Regularization strategies like dropout further enhanced the model’s generalizability, as evidenced by consistent validation metrics.

The sheer size of 3D MRI scans composed of numerous 2D slices stacked together to form a detailed 3D representation, makes them substantially larger than single 2D images. This increased size necessitates more computational resources and leads to longer processing times, slowing down training and escalating costs. Deep learning models that process 3D data are particularly resource-intensive compared to their 2D counterparts. Additionally, the substantial storage demands of 3D MRI data further complicate their usage. Using 2D slices instead of full 3D images can be an effective alternative to address these challenges. Training DL models with selected 2D slices that capture the most relevant information makes it possible to reduce computational and storage demands, making the process more feasible, particularly when resources are limited.

As stated previously, in this proposed method, each 3D sMRI is re-sliced into 2D images. This approach generates numerous slices; however, not all of them are equally informative. Some slices may primarily contain noise, while others are rich in valuable information. To optimize the training and testing of the studied model, the most informative slices selectively were extracted.

### 4.3. Limitations

Despite its promising results, the study has several limitations. The reliance on 2D slices, while computationally efficient, may result in the loss of volumetric information inherent in 3D MRI data. This could explain the slight decrease in performance for clinically similar stages such as EMCI and LMCI. Additionally, the datasets used (ADNI and OASIS) primarily consist of participants from controlled research settings, which may limit the generalizability of the model to diverse populations and real-world clinical data.

### 4.4. Clinical Implications

The proposed model offers significant potential for clinical application. Its ability to accurately classify AD stages supports early diagnosis and targeted intervention, particularly in distinguishing between EMCI and LMCI. The high F1-scores achieved for all classes (e.g., 0.99 for NC and EMCI, 0.94 for AD, and 0.98 for LMCI) confirm its reliability in balancing precision and recall. These metric values are critical in medical diagnostics to minimize misclassifications and improve patient outcomes.

### 4.5. Future Directions

To further enhance the model’s diagnostic utility, future research should explore the integration of multimodal data, such as PET imaging and CSF biomarkers, alongside MRI data. Employing 3D convolutional networks or hybrid models that combine 2D and 3D features may capture more nuanced structural changes. Expanding the dataset to include diverse imaging protocols and populations would improve generalizability. In addition, deploying the model in real-world clinical workflows could provide insights into its practical applicability and areas for refinement.

## 5. Conclusions

This study uses the basic model of DenseNet 169, as an advanced and practical network, for image classification of AD. By modifying this model for better learning in a shorter time and using transfer learning, the two-dimensional MR images considered the largest area of the brain (including all three parts of white and gray matter and CSF) an accuracy of 99.7% for the separation of Alzheimer’s classes of AD, EMCI, LMCI, and the NC control group was achieved.

The most important of this high accuracy is in separating the two groups of EMCI and LMCI, which were not distinguished in the literature, and considered both to be the same.

As a result, accurate diagnosis of EMCI and LMCI can lead to early prediction of AD (if EMCI is diagnosed) or prevention and slowing down of AD before its progress (if LMCI is diagnosed correctly). Also, the use of 2D images and comparing the results of this study with previous similar works that have been conducted using 3D images, shows it is possible to achieve great results by using a smaller volume of images and as a result, reducing time and requiring less powerful processor systems. Future research should aim to integrate multimodal data, employ advanced network architectures, and expand datasets to enhance the proposed model.

## Figures and Tables

**Figure 1 diagnostics-15-00360-f001:**
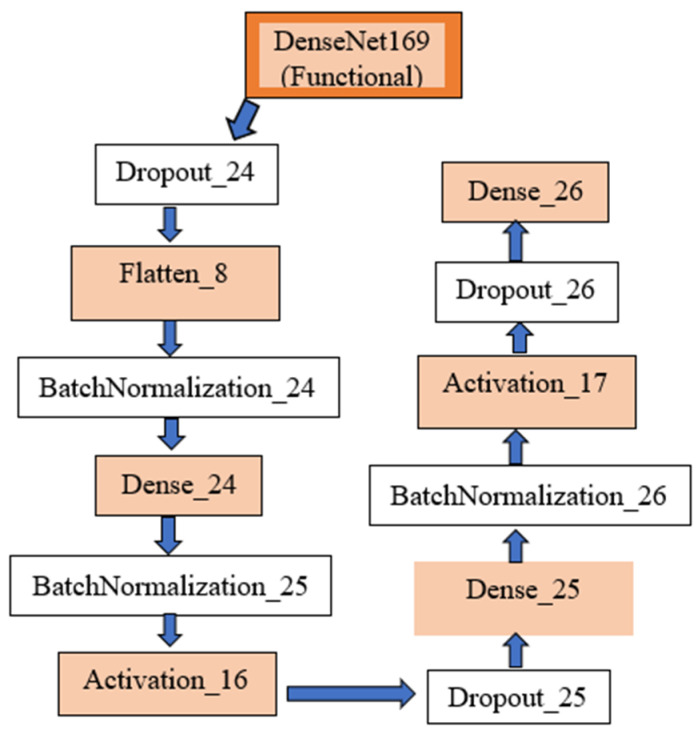
A diagram of the model architecture.

**Figure 2 diagnostics-15-00360-f002:**
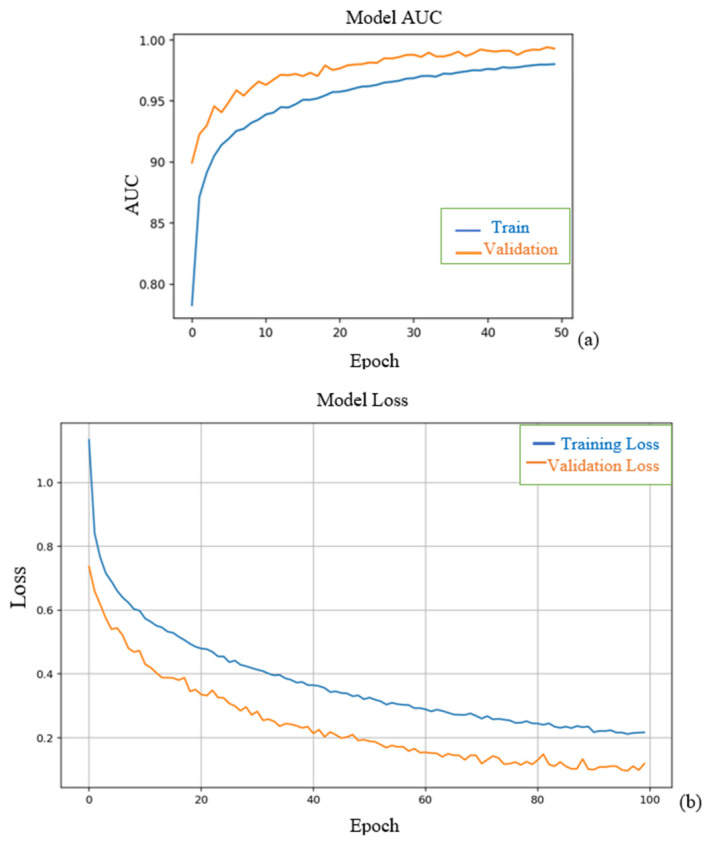
(**a**) A graphical representation of the training and validation loss curves. (**b**) The AUC curves for training and validation datasets.

**Figure 3 diagnostics-15-00360-f003:**
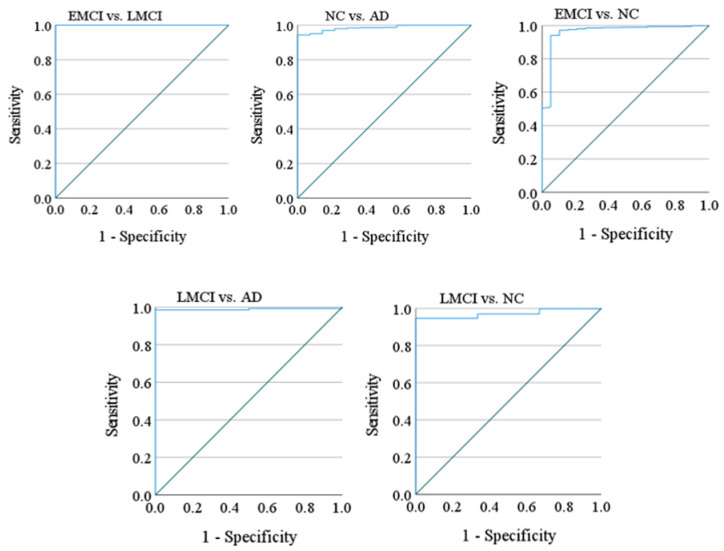
The ROC curve for the binary distinction of Alzheimer’s disease classes.

**Figure 4 diagnostics-15-00360-f004:**
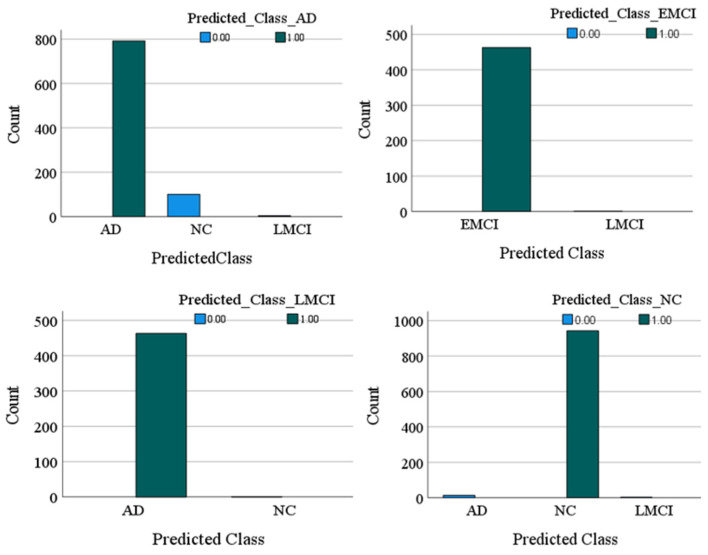
The prediction graph of each class compared to other classes.

**Table 1 diagnostics-15-00360-t001:** Demographic and clinical information (mean ± standard deviation) of the studied ADNI subjects.

Diagnosis	Age	Gender (M/F)	MMSE	CDR
AD	75.9 ± 6.8	47/51	23.2 ± 1.8	0.8 ± 0.2
EMCI	75.2 ± 7.1	48/49	26.9 ± 1.8	0.3 ± 0.1
LMCI	75.9 ± 4.8	48/49	25.2 ± 1.6	0.5 ± 0.1
NC	75.1 ± 7.6	48/49	29.2 ± 1.0	0 ± 0

AD: Alzheimer’s disease; EMCI: early mild cognitive impairment; LMCI: late mild cognitive impairment; NC: normal control; M: male; F: female; MMSE: mini-mental state examination; CDR: clinical dementia rating.

**Table 2 diagnostics-15-00360-t002:** The AUC values for predicting four classes and binary classification.

Class	AUC	Std. Error ^a^	Asymptotic Sig. ^b^	Lower Bound	Upper Bound
NC vs. AD	0.985	0.005	0.000	0.974	0.995
EMCI vs. NC	0.961	0.025	0.000	0.912	1.000
LMCI vs. NC	0.951	0.333	0.000	0.885	1.000
LMCI vs. AD	0.989	0.004	0.001	0.982	0.997
EMCI vs. LMCI	1.000	0.000	0.084	1.000	1.000

^a^ Under the nonparametric assumption, ^b^ null hypothesis: true area = 0.5. Normal control (NC), late mild cognitive impairment (LMCI), Alzheimer’s disease (AD), and early mild cognitive impairment (EMCI) by comparing binary classes.

**Table 3 diagnostics-15-00360-t003:** Confusion matrix and F1-scores of four classes.

Actual\Predicted	AD	CN	EMCI	LMCI	F1-Scores
AD	792	100	0	4	0.94
NC	14	943	0	3	0.99
EMCI	0	0	463	1	0.99
LMCI	4	14	1	877	0.98

**Table 4 diagnostics-15-00360-t004:** Comparison of classification performance with state-of-art studies.

Study	Dataset	Classification Tasks	Input	Transfer Learning	Accuracy%
[3]	ADNI	NC, AD, MCI	3D sMRI	no	88.9
[30]	ADNI	NC, AD, MCI	2D sMRI	no	62.0
[31]	ADNI	NC, AD, MCI	3D sMRI	yes	82.4
[27]	ADNI, OASIS	NC, AD, EMCI, LMCI	2D sMRI	yes	91.45
The present Study	ADNI, OASIS	NC, AD, EMCI, LMCI	2D sMRI	yes	99.7

## Data Availability

The data presented in this study are available on request from the corresponding author.

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
