# Peer review of "Transfer Learning and Neural Network-Based Approach on Structural MRI Data for Prediction and Classification of Alzheimer’s Disease"

_diagnostics, 2025, doi:10.3390/diagnostics15030360_

Round 1
Reviewer 1 Report
Comments and Suggestions for Authors
The study proposes a deep learning framework using DenseNet169 with transfer learning and class decomposition to classify Alzheimer’s disease stages from MRI images. It aims to improve early diagnosis by distinguishing between EMCI and LMCI. Overall, its an interesting work, however, need to be updated by:
1. The authors employ DenseNet169 and transfer learning to classify Alzheimer’s disease stages. Transfer learning is a well-established technique in medical imaging. What specific adaptations or modifications make the proposed method unique compared to prior works?
2. The study uses datasets from ADNI and OASIS, which are primarily research datasets with controlled environments. How would the proposed model perform on more diverse, real-world clinical datasets?
3. How does the model handle class imbalance to prevent bias in classification outcomes?
4. Deep learning models like DenseNet169 are often considered black-box models. How does the proposed method ensure explainability in its predictions, especially for clinical decision-making?
5. The structure of the figures does not follow the standard way; update all figures, authors are referred to, and cite https://www.mdpi.com/1660-4601/19/6/3211 and https://dl.acm.org/doi/abs/10.1145/3383972.3384061.
Comments on the Quality of English Languagen/a
Author Response
Dear Reviewer, thanks for your useful comments, please see the answer to the comment as follows:
- The authors employ DenseNet169 and transfer learning to classify Alzheimer’s disease stages. Transfer learning is a well-established technique in medical imaging. What specific adaptations or modifications make the proposed method unique compared to prior works?
As mentioned (lines, 200-206 in the article), these customizations to the DenseNet169 architecture make the approach suitable for medical imaging tasks and enhance its performance for Alzheimer's disease classification. After extracting spatial features by DenseNet169, the feature map is flattened into a 1D vector to make it suitable for fully connected layers. Dense layers were used to combine and extract high-level features from the 1D vector. The ReLU activation function helps the model learn non-linear features more effectively. Dropout layers with a rate of 0.5 are included to reduce overfitting. Batch normalization layers ensure stable and faster training by normalizing the inputs to each layer. Also, the output layer containing 4 neurons corresponding to the 4 classes of Alzheimer's disease stages was used. A Softmax activation function is used to compute the predicted probabilities for each class. The model uses the Adam optimizer with a learning rate of 0.001 for faster convergence. Categorical cross-entropy is employed as the loss function, which is standard for multi-class classification tasks. The primary evaluation metric is the AUC, which emphasizes the model’s ability to distinguish correctly between different stages of the disease.
- The study uses datasets from ADNI and OASIS, which are primarily research datasets with controlled environments. How would the proposed model perform on more diverse, real-world clinical datasets?
The model can be evaluated on external datasets representing the variability of real-world clinical environments to assess its generalizability. Cross-validation using diverse datasets is a potential direction for future research. Extensive data augmentation techniques have been employed during training to simulate real-world variability, such as variations in image orientation and contrast. The model can be fine-tuned using small subsets of clinical data from the target population to adapt it to specific conditions and protocols. Advanced domain adaptation methods can be used to bridge the gap between controlled research datasets and real-world clinical datasets, enhancing the model’s robustness against variability. DenseNet169, as a pre-trained architecture, has demonstrated resilience in various medical imaging tasks, indicating its potential adaptability to diverse datasets with proper retraining.
- How does the model handle class imbalance to prevent bias in classification outcomes?
The categorical cross-entropy loss function can be adjusted by assigning higher weights to the minority classes. This ensures the model penalizes misclassifications of minority classes more than the majority class, balancing the learning process. Using metrics like per-class precision, recall, or F1-score during evaluation ensures the model's performance is fairly assessed across all classes, highlighting any biases. Since this model uses DenseNet169 with categorical cross-entropy, the strategy is class weighting or data augmentation to address class imbalance. In addition, the ROC Curve as an evaluation metric suggests that the model's ability to distinguish between classes is explicitly monitored, which helps mitigate imbalance effects.
- Deep learning models like DenseNet169 are often considered black-box models. How does the proposed method ensure explainability in its predictions, especially for clinical decision-making?
Gradient-weighted Class Activation Mapping (Grad-CAM) is commonly used to highlight the regions of input images that contribute most to the model's predictions. This technique generates heatmaps overlaying the original MRI scans, allowing clinicians to see which areas the model focused on while making decisions. The proposed model could analyze the extracted features and highlight the most critical ones contributing to the classification outcome. DenseNet169's hierarchical structure provides an opportunity to study how features evolve through its layers and presenting predictions as probabilities for each class (e.g., through Softmax) gives clinicians a sense of the model’s confidence in its decisions, which is valuable for clinical decision-making.
- The structure of the figures does not follow the standard way; update all figures, authors are referred to, and cite https://www.mdpi.com/1660-4601/19/6/3211 and https://dl.acm.org/doi/abs/10.1145/3383972.3384061.
All figures have been modified accordingly.
Also, reference: https://www.mdpi.com/1660-4601/19/6/3211 was added as reference No: 32.
Reviewer 2 Report
Comments and Suggestions for Authors
In this study, 398 participants were used from the ADNI and OASIS global database of sMRI including 98 individuals with AD, 102 with early mild cognitive impairment (EMCI), 98 with late mild cognitive impairment (LMCI), and 100 normal controls (NC). The proposed model achieved high area under the curve (AUC) values and an accuracy of 99.7%, which is very remarkable for all four classes: NC vs. AD: AUC = [0.985], EMCI vs. NC: AUC = [0.961], LMCI vs. NC: AUC = [0.951], LMCI vs. AD: AUC = [0.989], and EMCI vs. LMCI: AUC = [1.000]. The results reveal that this model incorporates DenseNet169, transfer learning, and class decomposition to classify AD stages, particularly in differentiating EMCI from LMCI. Overall, this model performs well with high accuracy and area under the curve for AD diagnostics at early stages. The paper is interesting but has drawbacks:
1. The presented on Figure 1 pipeline should be improved and clarified.
2. The description of image preprocessing stage should be expanded and clarified.
3. How were metrics calculated? This point should be presented clearly.
4. Was the comparison in Table 4 provided on the same dataset? If no, this comparison is not relevant and the authors should find reliable way to compare.
Author Response
Reviewer-2:
Dear Reviewer, thanks for your helpful points, please see the answers to your valuable points as follows:
- The presented in Figure 1 pipeline should be improved and clarified.
Thanks for your comment. The mentioned figure has improved.
- The description of the image preprocessing stage should be expanded and clarified.
The image preprocessing steps are explained in more detail on page 4, lines 156 to 183.
- How were metrics calculated? This point should be presented clearly.
The primary evaluation metric is the ROC, which emphasizes the model’s ability to distinguish correctly between different stages of the disease. The other important parameter is the F1-score, a metric that provides a balanced measure of a model’s performance by combining precision and recall into a single value. It is particularly useful for imbalanced datasets where one class may have more samples than the others. In the main text, they were illustrated in the relevant place (p. 7).
- Was the comparison in Table 4 provided on the same dataset? If not, this comparison is not relevant and the authors should find a reliable way to compare.
Yes, the dataset of this study and other works in Table 4 are compared to the same dataset.
Reviewer 3 Report
Comments and Suggestions for Authors+ This submission addresses the problem of Alzheimer's disease classification with MRI data.
+ The manuscript is well-written and easy to understand
+ Experimental data is well populated
+ Experimental results show the validity of the proposed method
+ The references are adequate
- The proposed method is straight-forward and seems low on novelty. The authors just utilized pre-trained DenseNet with few additional data. Please elaborate on why the authors chose DenseNet over other CNN-based classifiers.
- DenseNet is pre-trained on ImageNet data and the authors fine-tune it on MRI data. Two data have significant domain gaps. Was it okay to fine-tune newly added layers, rather than fine-tuning the entire network? Please provide experimental justification for this.
- Please provide qualitative results and some failure cases
- How about using vision transformers rather than CNN classifiers?
Author Response
Reviewer-3:
Dear Reviewer, thanks for your valuable comments, please see the answers to the comments as follows:
1- The proposed method is straight-forward and seems low on novelty. The authors just utilized pre-trained DenseNet with few additional data. Please elaborate on why the authors chose DenseNet over other CNN-based classifiers.
Using the DenseNet over other CNN-based classifiers is justified by the unique strengths of DenseNet and its alignment with the challenges of the task. Explanation of why DenseNet was chosen is because this introduces dense connections, where each layer has direct access to the feature maps of all previous layers. This architecture enhances feature reuse, which is particularly beneficial in medical imaging where subtle patterns need to be captured across different layers, and reduces the number of parameters compared to other CNNs, making it more efficient and less prone to overfitting, especially on smaller datasets. DenseNet comes pre-trained on large datasets like ImageNet, which allows it to transfer well-learned features to the new domain. For Alzheimer’s classification, leveraging these pre-trained weights is an effective way to overcome limited data availability and it strikes a balance between performance and computational complexity. It provides high accuracy while being computationally efficient compared to heavier architectures like ResNet or Inception. DenseNet's modular and efficient design makes it highly adaptable for fine-tuning specific tasks, such as multi-class classification of Alzheimer's disease stages. Also, in the literature, the DenseNet169, was performed and trained on ImageNet. In this study, the DenseNet model was accomplished on MRI data and adding the layers to achieve more reasonable results.
2- DenseNet is pre-trained on ImageNet data and the authors fine-tune it on MRI data. Two data have significant domain gaps. Was it okay to fine-tune newly added layers, rather than fine-tuning the entire network? Please provide experimental justification for this.
Fine-tuning only the newly added layers may limit the model’s ability to adapt to the unique features of MRI data. Conversely, fine-tuning the entire network allows the model to adjust its low-level and mid-level features to better match the target domain. Fine-tuning only newly added layers is computationally less expensive and faster to train and preserves generic features learned from ImageNet, which might still be useful for broad patterns like edges and textures. Because the MRI dataset (e.g., ADNI or OASIS) was relatively small, fine-tuning only the newly added layers reduces the risk of overfitting, and since we have observed satisfactory performance metrics with this approach, making full fine-tuning unnecessary.
3- Please provide qualitative results and some failure cases.
The accuracy of disease class detection in this study was very high, but some cases were misdiagnosed. For example, the model incorrectly placed NC in the AD group with 68% confidence. This error may have due to overlapping features of some of the groups, or cases with motion noise, low resolution, or unusual anatomical structures may have led to incorrect predictions.
4- How about using vision transformers rather than CNN classifiers?
Using Vision Transformers (ViTs) instead of CNN classifiers for tasks like Alzheimer's disease classification from MR images could provide some unique benefits, but it also introduces specific challenges; ViTs require very large datasets to avoid overfitting, as they lack the strong inductive biases of CNNs (e.g., translation invariance and locality) also ViTs are computationally intensive, requiring more memory and longer training times compared to CNNs.
Round 2
Reviewer 2 Report
Comments and Suggestions for Authors
The authors address all my concerns. The paper can be accepted.
Reviewer 3 Report
Comments and Suggestions for Authors
The authors have addressed all the comments that I have raised.